# Effect of automated versus conventional ventilation on mechanical power of ventilation—A randomized crossover clinical trial

**Laura A. Buiteman-Kruizinga**[1,2]*, **Ary Serpa Neto**[2,3,4,5,6], **Michela Botta**[2], **Stephanie S. List**[7], **Ben H. de Boer**[7], **Patricia van Velzen**[7], **Philipp Karl Bühler**[8], **Pedro D. Wendel Garcia**[8], **Marcus J. Schultz**[2,9,10,11], **Pim L. J. van der Heiden**[1], **Frederique Paulus**[2,12], for the INTELLiPOWER–investigators[¶]

1 Department of Intensive Care, Reinier de Graaf Hospital, Delft, the Netherlands, 2 Department of Intensive Care, Amsterdam University Medical Centers 'Location AMC', Amsterdam, the Netherlands, 3 Australian and New Zealand Intensive Care–Research Centre (ANZIC–RC), Monash University, Melbourne, Australia, 4 Department of Intensive Care, Austin Hospital, Melbourne, Australia, 5 Department of Critical Care, University of Melbourne, Melbourne, Australia, 6 Department of Critical Care Medicine, Hospital Israelita Albert Einstein, São Paulo, Brazil, 7 Department of Intensive Care, Dijklander Hospital 'Location Hoorn', Hoorn, the Netherlands, 8 Institute of Intensive Care Medicine, University Hospital Zürich, Zürich, Switzerland, 9 Mahidol–Oxford Tropical Medicine Research Unit (MORU), Mahidol University, Bangkok, Thailand, 10 Nuffield Department of Medicine, University of Oxford, Oxford, United Kingdom, 11 Department of Anesthesia, General Intensive Care and Pain Management, Medical University Wien, Vienna, Austria, 12 ACHIEVE, Centre of Applied Research, Faculty of Health, Amsterdam University of Applied Sciences, Amsterdam, The Netherlands

¶ INTELLiPOWER: 'The Effect of Closed–loop versus Conventional Ventilation on Mechanical Power' and a complete list of collaborative authors is provided in the Acknowledgments.
* l.kruizinga@rdgg.nl

**Data Availability Statement:** The data and R scripts are available from the online public

## Abstract

### Introduction

Mechanical power of ventilation, a summary parameter reflecting the energy transferred from the ventilator to the respiratory system, has associations with outcomes. INTELLi-VENT–Adaptive Support Ventilation is an automated ventilation mode that changes ventilator settings according to algorithms that target a low work–and force of breathing. The study aims to compare mechanical power between automated ventilation by means of INTELLi-VENT–Adaptive Support Ventilation and conventional ventilation in critically ill patients.

### Materials and methods

International, multicenter, randomized crossover clinical trial in patients that were expected to need invasive ventilation > 24 hours. Patients were randomly assigned to start with a 3–hour period of automated ventilation or conventional ventilation after which the alternate ventilation mode was selected. The primary outcome was mechanical power in passive and active patients; secondary outcomes included key ventilator settings and ventilatory parameters that affect mechanical power.

repository figshare (https://figshare.com/s/
5765819d443d6e91ab1a).

**Funding:** The author(s) received no specific
funding for this work.

**Competing interests:** LBK received fees from
Hamilton Medical for lecturing. MJS was part-time
employed as a team leader of Research and New
Technologies at Hamilton Medical from January
2022 till January 2023. The other authors declare
no conflicts of interest.

## Results

A total of 96 patients were randomized. Median mechanical power was not different
between automated and conventional ventilation (15.8 [11.5–21.0] versus 16.1 [10.9–22.6]
J/min; mean difference –0.44 (95%–CI –1.17 to 0.29) J/min; $P = 0.24$). Subgroup analyses
showed that mechanical power was lower with automated ventilation in passive patients,
16.9 [12.5–22.1] versus 19.0 [14.1–25.0] J/min; mean difference –1.76 (95%–CI –2.47 to –
10.34J/min; $P < 0.01$), and not in active patients (14.6 [11.0–20.3] vs 14.1 [10.1–21.3] J/min;
mean difference 0.81 (95%–CI –2.13 to 0.49) J/min; $P = 0.23$).

## Conclusions

In this cohort of unselected critically ill invasively ventilated patients, automated ventilation
by means of INTELLiVENT–Adaptive Support Ventilation did not reduce mechanical power.
A reduction in mechanical power was only seen in passive patients.

## Study registration

Clinicaltrials.gov (study identifier NCT04827927), April 1, 2021

## URL of trial registry record

https://clinicaltrials.gov/study/NCT04827927?term=intellipower&rank=1

## Introduction

The mechanical power of ventilation (MP) is the amount of energy per time transferred from
the ventilator to the respiratory system [1]. This energy is used to overcome the resistance of
the airway ($R_{AW}$) and the compliance of the respiratory system ($C_{RS}$) to deliver a breath [2–4].
Part of this energy, however, can act directly on lung tissue, including the endothelial and epi-
thelial cells and the lung skeleton where it may cause injury [5, 6]. MP has been shown to have
associations with important patient–centered outcomes in various types of critically ill patients
under invasive ventilation [7–11].

MP is a summary parameter that includes the components known to play a role in ventila-
tor–induced lung injury (VILI) [1], i.e., tidal volume ($V_T$) [12], plateau pressure (Pplat) and
driving pressure (ΔP), and respiratory rate (RR). Several equations have been proposed for cal-
culating MP at the bedside [1, 13–17]. With so many ventilation variables to adjust, some of
which even with an opposite or non–intuitive impact on MP, it could be difficult to target a
low MP, because it is uncertain which setting to prioritize.

Closed–loop, or automated ventilation modes are increasingly available for use in critically
ill invasively ventilated patients [18]. INTELLiVENT–Adaptive Support Ventilation (ASV) is
one automated mode that sets and adjusts $V_T$, RR, positive end–expiratory pressure (PEEP)
and the fraction of inspired oxygen ($FiO_2$), after inserting gender and height, target ranges for
the end–tidal $CO_2$ ($etCO_2$) and pulse oximetry ($SpO_2$), and limits for maximum airway pres-
sure and PEEP into the ventilator. INTELLiVENT–ASV then changes ventilator settings based
on algorithms that target a low work–and force of breathing [19, 20], including settings that
may affect MP [21–24].

We performed a randomized crossover clinical trial, named 'The Effect of Closed–loop ver-
sus Conventional Ventilation on Mechanical Power' (INTELLiPOWER) to test the hypothesis
that automated ventilation results in less MP when compared to conventional ventilation.

## Materials and methods

### Study design

'INTELLiPOWER' was an investigator–initiated, international, multicenter, randomized crossover clinical trial. This study was conducted in the intensive care units (ICUs) at the Reinier de Graaf Hospital in Delft, the Dijklander hospital 'location Hoorn' in Hoorn, the Amsterdam University Medical Centers, 'location AMC', Amsterdam, the Netherlands, and the University Hospital Zürich, Zürich, Switzerland, between July 3, 2021 and April 15, 2023.

The study protocol of INTELLiPOWER was first approved by the institutional review board of the AMC (2020_317#B2021122, February 26, 2021). Following this approval, patient recruitment began in the Netherlands. The protocol was additionally approved by the Cantonal Ethics Commission Zürich (Swissethics 2023–D0012, February 28, 2023). Following the additional approval, patient recruitment commenced in Switzerland.

Written informed consent was obtained from the legal representative of each patient before inclusion and randomization, and the study underwent external monitoring by a notified body. INTELLiPOWER was registered at clinicaltrials.gov (study identifier NCT04827927, registered April 1 2021). CONSORT reporting guidelines were used [25]. A statistical analysis plan was written and finalized before cleaning and closing of the database. Further details can be found in the S1 File.

### Patients

Patients were eligible for participation if: (1) aged $\geq$ 18 years; (2) expected to need invasive ventilation for $>$ 24 hours; (3) having the ability to randomize as soon as possible, but always $<$ 48 hours after start of invasive ventilation in the ICU; and (4) ventilated with a ventilator that was able to provide the automated ventilation mode of interest (i.e., a Hamilton ventilator [Hamilton Medical, Bonaduz, Switzerland]). Patients receiving invasive ventilation though a tracheostomy cannula were excluded. We also excluded patients with a body mass index $>$ 40, and patients with any contraindication for use of the here tested automated ventilation mode.

### Randomization and masking

Patients were randomly assigned in a 1:1 ratio to start with automated ventilation or conventional ventilation for 3 hours, after which the alternate ventilation mode was applied. Between each ventilation mode, there was a 30–minutes wash out period. A dedicated, password protected, web–based randomization system (SSL–encrypted website, Castor Electronic Data Capture, Amsterdam, the Netherlands) was used by the local investigators for randomization using random block sizes of 4 or 6 patients, stratified per center. Doctors, ventilation practitioners and nurses taking care of patients could not be blinded because of the nature of the intervention. The investigators analyzing the data, however, remained blinded for the allocated ventilation mode at all times.

### Study interventions

The predecessor of INTELLiVENT–ASV, in short ASV, is a ventilation mode that adapts to patients' respiratory mechanics, providing settings as needed. It simplifies the ventilatory management process by adjusting $V_T$ and RR automatically, based on the patients' conditions. INTELLiVENT–ASV incorporates advanced algorithms and real–time monitoring to further optimize ventilation parameters and support patient breathing. Herein, PEEP and $FiO_2$ are adjusted with the PEEP controller and $FiO_2$ controller, and $V_T$ and RR are further adjusted by

a minute ventilation controller. With INTELLiVENT–ASV, the ventilation and oxygenation targets are chosen via ranges for $etCO_2$ and $SpO_2$ readings. The ventilator then sets and adjusts breath–by–breath the size of $V_T$ in conjunction with the RR, and the level of PEEP and $FiO_2$. Doctors, ventilation practitioners and nurses were extensively trained and qualified in using the ventilators used for this study, and were experienced users of the here tested automated ventilation mode INTELLiVENT–ASV.

After intubation, patients received conventional ventilation by means of pressure controlled or pressure support ventilation, or automated ventilation, according to the local ventilation protocol, until patients were randomized to start either with conventional ventilation or automated ventilation. Similar $etCO_2$ and $SpO_2$ were used during the crossover phases of the study. Sedation depth was not changed, and patients were not subjected to other activities, such as daily care like washing, physiotherapy or airway interventions like airway suctioning, unless strictly necessary.

With automated ventilation, the attending ICU doctor, nurse or ventilation practitioner set the $etCO_2$ and $SpO_2$ target ranges using the same goals as before randomization. The controllers for minute volume, PEEP and $FiO_2$ were all activated, so that the ventilator software can adjust $V_T$, RR, PEEP and $FiO_2$ according to the algorithms that continuously target a low work of breathing and a low force of breathing [19, 20]. The lower PEEP limit was 5 cm $H_2O$, the higher PEEP limit varied from 10 to 15 cm $H_2O$, depending on the patients' lung or clinical conditions, and according to the local protocol for ventilation.

With conventional ventilation, the attending ICU doctor, nurse or ventilation practitioner set the ventilator using the same goals for $etCO_2$ and $SpO_2$ as before randomization. Lung–protective settings were advised to be used, as indicated in the local ventilation protocol, including using low $V_T$ (6–8 ml/kg predicted body weight [PBW]), low maximum airway pressure ($P_{max}$, 30 cm $H_2O$), and a low $\Delta P$ ($< 15$ cm $H_2O$); PEEP was set according to a lower PEEP/$FiO_2$ table [26], wherein the lowest allowed PEEP was 5 cm $H_2O$, and $FiO_2$ was adjusted to maintain the $SpO_2$ within range.

It is worth noting that the transition between modes did not involve changing the ventilator itself, but rather only switching the ventilation mode. Patients were ventilated with the same ventilator throughout, transitioning between modes while receiving ventilation.

## Data collection

Ventilation parameters were collected at 12 consecutive time points: six time points per each ventilation mode. Inspiratory and expiratory holds were performed every 30 minutes in passively ventilated patients to measure the static ventilation pressures and to determine the absence of spontaneously breathing activity. Researchers were extensively trained and experienced in the performance of these holds and measurements [27]. At each time point, we collected the inspired and expired $V_T$ ($V_{Ti}$ and $V_{Te}$, mL), set and measured RR (breaths per minute), maximum airway pressure (Pmax, cm $H_2O$), Pplat (cm $H_2O$), set and total PEEP (cm $H_2O$), set inspiratory pressure (Pinsp, cm $H_2O$) and inspiratory time (Tinsp, sec). We also collected the rise time (Tslope, sec), inspiratory flow (L/min), $FiO_2$ (%), $etCO_2$ (kPa) and $SpO_2$ (%). When each phase lasted >1 hour and patient was stable, an arterial blood gas analysis was performed. A follow–up was performed at day 28.

## Outcomes

The primary outcome was MP in the entire cohort. Secondary outcomes were other ventilatory parameters that have an association with MP and are affected by the automated mode tested in this study, i.e., $V_T$, $\Delta P$, RR, Pmax and Pplat, PEEP and Pinsp.

## Sample size calculation

Considering a mean MP of 24.2 J/min and a standard deviation of 10.5 J/min with conventional ventilation, 96 patients would be needed to demonstrate a 15% relative reduction (3.5 J/min) with 90% power and 5% of alpha. This calculation was based on the results of a non–randomized parallel group study that showed a reduction in MP from 24.2 (± 9.2) to 18.0 (± 11.4) J/min [28].

## Statistical analysis

Data are expressed in numbers and proportions for categorical variables and medians with interquartile ranges or means with the standard deviation for continuous variables. Proportions are compared using the chi–squared test or Fisher exact test as required by variable distribution; continuous variables are compared using paired T–test or Wilcoxon signed–rank test, where appropriate. Effects are reported with a 95%–confidence interval (95%–CI). Ventilation data are reported in tables and visualized in cumulative distribution plots and spider charts.

For the primary analysis, we first compared MP with automated ventilation to conventional ventilation in all patients, and then in two subgroups, i.e., patients that were passive and patients that were actively breathing at all time points. Herein, MP between ventilation modes was compared using a mixed–effect generalized linear model with Gaussian distribution, including time of measurements as a continuous variable, treatment group (automated or conventional), period (1 or 2) and sequence (automated to conventional, or conventional to automated) as fixed effects, and patients and centers as random effect to account for repeated measurements. The models were checked to meet the Gauss–Markov assumptions (S1–S3 Figs).

We used the following equation to calculate MP [14], one that has been proposed to be used in patients under pressure controlled ventilation and $\Delta P$ [29]:

$$MP(J/min) = 0.098*RR*V_T*(Pinsp + PEEP); \qquad [Eq1]$$

$$\Delta P_{dynamic}(cm\ H_2O) = Pmax - PEEP_{set}; \qquad [Eq2]$$

$$\Delta P_{static} = Pplat - PEEP_{tot}. \qquad [Eq3]$$

Other endpoints, i.e., RR, $V_T$ and additional ventilatory parameters were compared using the same model as for the primary endpoint. We constructed cumulative distribution graphs to visualize cumulative distribution frequencies of MP, wherein vertical dotted lines represent the broadly accepted safety cutoffs for MP, and horizontal dotted lines show the respective proportion of patients reaching that cutoff. Next, we constructed spider charts to visualize the proportions of patients with MP > 17 J/min, $V_T$ > 8 ml/kg PBW, flow > 45 L/min, RR > 16/min, Pplat or Pmax > 20 cm $H_2O$, and $\Delta P$ > 12 cm $H_2O$. These cutoffs were chosen based on what was used in previous studies [7, 12, 29]. Distribution graphs and spider charts were constructed for passive and active patients separately.

We performed three posthoc analyses. We compared MP between the automated and conventional ventilation in patients who received conventional ventilation before randomization, separately from patients who received automated ventilation before randomization; we compared MP between the automated and conventional ventilation in patients with a low versus a high $C_{RS}$, using the median baseline of 37.3 mL/cmH$_2$O as a cutoff; and we compared the proportion of time points with MP > 17 J/min and MP ≤ 17 J/min between the automated and conventional ventilation, and performed a time–weighted average comparison.

All analyses followed the intention–to–treat principle. Missing data occurred completely at random. The proportion of missing data was very low (< 0.06%) and we made no assumptions for missing data. A $P$–value < 0.05 was considered significant. All analyses were performed with R version 4.0.3 (R Foundation, Vienna, Austria).

## Results

### Patients

96 patients were included, of which 50 patients started with automated ventilation and 46 patients with conventional ventilation, and then crossed over to the alternate mode (Fig 1). The majority of patients were male and admitted for a medical reason (Table 1). The main reasons for invasive ventilation were acute hypoxemic respiratory failure due to pneumonia or cardiac arrest. Most patients were randomized within 24 hours after start of invasive ventilation. In the automated–conventional sequence group, the APACHE IV score was lower and more patients had cardiovascular disease as comorbidity.

### Mechanical power

In the entire cohort, median MP was not different between automated and conventional ventilation (15.8 [11.5–21.0] vs 16.1 [10.9–22.6] J/min; mean difference–0.44 (95%–CI –1.17 to 0.29) J/min; $P$ = 0.24).).

### Subgroup analyses

A lower median MP with automated ventilation was found in passive patients, 16.9 [12.5–22.1] versus 19.0 [14.1–25.0] J/min; mean difference –1.76 (95%–CI –2.47 to –10.34J/min; $P$ < 0.01), and not in active patients (14.6 [11.0–20.3] versus 14.1 [10.1–21.3] J/min; mean difference 0.81 (95%–CI –2.13 to 0.49) J/min; $P$ = 0.23). (Table 2 and Fig 2).

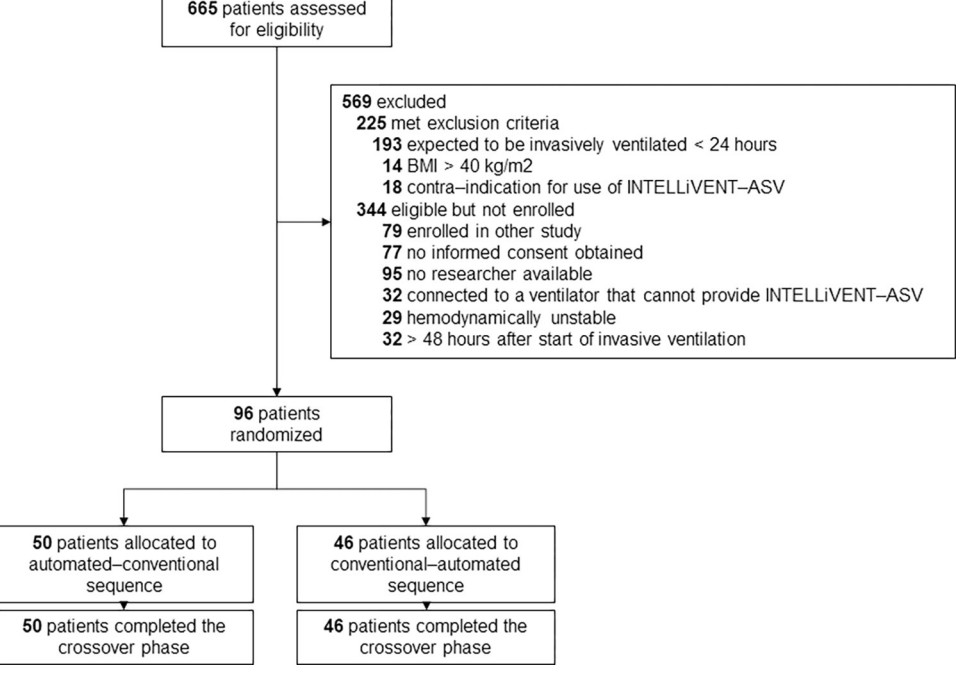

**Fig 1. Consort diagram showing the flow of patients.**

**Table 1. Baseline characteristics and outcomes.**

| | all (n = 96) | automated–conventional sequence (n = 50) | conventional–automated sequence (n = 46) |
|---|---|---|---|
| Gender (male) | 71 (74) | 39 (78.0) | 32 (70.0) |
| Age (years) | 66 [58–73] | 65 [58–75] | 67 [59–72] |
| Height (cm) | 174 [169–180] | 177 [167–184] | 172 [170–176] |
| Weight (kg) | 80.2 [72.0–91.8] | 83.0 [75.0–95.0] | 75.0 [68.8–85.8] |
| PBW (kg) | 69.7 [62.4–75.1] | 72.4 [62.4–78.8] | 66.5 [62.6–71.5] |
| BMI (kg/m$^2$) | 26.2 [23.3–29.4] | 26.9 [24.1–29.6] | 26.0 [23.0–29.3] |
| APACHE IV score | 74 [52–96] | 66 [40–94] | 77 [62–98] |
| Comorbidities | | | |
| Cardiovascular disease | 44 (45.8) | 25 (50.0) | 19 (41.3) |
| COPD | 8 (8.3) | 2 (4.0) | 6 (13.0) |
| Neurological condition | 10 (10.4) | 5 (10.0) | 5 (10.9) |
| Reason for ICU admission | | | |
| Medical condition | 88 (91.7) | 43 (86.0) | 45 (97.8) |
| Emergency surgery | 7 (7.3) | 6 (12.0) | 1 (2.2) |
| Elective surgery | 1 (1) | 1 (2.0) | 0 (0.0) |
| Reason for mechanical ventilation | | | |
| Respiratory failure | 65 (67.7) | 31 (62.0) | 34 (73.9) |
| Cardiac arrest | 15 (15.6) | 8 (16.0) | 7 (15.2) |
| Decreased consciousness | 11 (11.5) | 7 (14.0) | 4 (8.7) |
| Postoperative ventilation | 5 (5.2) | 4 (8.0) | 1 (2.2) |
| Causes of respiratory failure | | | |
| Pneumonia | 17 (17.7) | 8 (16.0) | 9 (19.6) |
| COVID–19 | 14 (14.6) | 7 (14.0) | 7 (15.2) |
| Obstructed airway | 6 (6.3) | 5 (10.0) | 1 (2.2) |
| Exacerbation COPD | 4 (4.2) | 1 (2.0) | 3 (6.5) |
| Causes of decreased consciousness | | | |
| Stroke | 2 (2.1) | 0 (0.0) | 2 (4.3) |
| Seizures | 2 (2.1) | 2 (4.0) | 0 (0.0) |
| Intoxication | 2 (2.1) | 0 (0.0) | 2 (4.3) |
| Neurotrauma | 2 (2.1) | 1 (2.0) | 1 (2.2) |
| Postoperative ventilation | | | |
| Gastro–intestinal bleeding | 4 (4.2) | 2 (4.0) | 2 (4.3) |
| Duration of ventilation before start of study, median [IQR] and mean (SD) (hours) | 17.0 [9.8–24.8] | 16.0 [9.0–21.0] | 18.5 [10.0–25.8] |
| | 17.4 (10.2) | 16.6 (10.0) | 18.3 (10.4) |
| Duration of ventilation (days) | 4.0 [3.0–7.0] | 4.0 [2.3–7.8] | 4.0 [3.0–6.0] |
| Alive at day 7 | 83 (86.4) | 43 (86.0) | 40 (87.0) |
| Alive at day 28 | 67 (69.8) | 34 (68.0) | 33 (71.7) |
| **Pre–randomization ventilation variables** | | | |
| Humidification | | | |
| Active | 80 (83.3) | 43 (86.0) | 37 (80.4) |
| Passive | 16 (16.7) | 7 (14.0) | 9 (19.6) |
| Ventilation mode | | | |
| automated | 62 (65) | 37 (74.0) | 25 (54.3) |

*(Continued)*

**Table 1.** (Continued)

| | all (n = 96) | automated–conventional sequence (n = 50) | conventional–automated sequence (n = 46) |
|---|---|---|---|
| conventional | 34 (35) | 13 (26.0) | 21 (45.7) |
| MP (J/min) | 16.6 [11.8–22.7] | 16.9 [15.6–22.6] | 14.2 [10.7–21.9] |
| $V_{Ti}$ (mL) | 467 [403–593] | 465 [403–612] | 471 [402–560] |
| $V_{Te}$ (mL) | 483 [399–600] | 494 [399–623] | 482 [398–562] |
| $V_T$ (ml/kg PBW) | 7.1 [5.7–8.2] | 7.1 [5.6–8.0] | 7.0 [5.8–8.5] |
| RR (breaths /minute) | 19 [15–23] | 19 [16–23] | 18 [14–22] |
| Minute volume (cm $H_2O$) | 8.8 [7.3–11.0] | 8.8 [7.7–10.9] | 8.8 [6.9–11.3] |
| Pmax (cm $H_2O$) | 22 [17–26] | 22 [18–25] | 22 [17–28] |
| PEEP, set (cm $H_2O$) | 8 [6–10] | 8 [5–10] | 8 [6–10] |
| Pinsp[+] (cm $H_2O$) | 13 [11–15] | 13 [11–15] | 14 [11–15] |
| PS[*] (cm $H_2O$) | 10 [5–12] | 11 [8–12] | 8 [5–12] |
| ΔP, dynamic (cm $H_2O$) | 13 [11–16] | 13 [11–16] | 13 [10–16] |
| Flow (L/min) | 43.3 [36.2–53.3] | 43.9 [34.9–53.0] | 42.3 [36.4–53.7] |
| Tinsp (sec) | 1.09 [0.80–1.20] | 1.08 [0.79–1.33] | 1.08 [0.83–1.18] |
| $FiO_2$ (%) | 34 [30–40] | 34 [30–38] | 35 [30–40] |
| $etCO_2$ (kPa) | 4.9 [4.4–5.4] | 4.9 [4.1–5.3] | 5.0 [4.5–5.5] |
| $SpO_2$ (%) | 95 [93–97] | 95 [93–97] | 95 [92–97] |
| $C_{RS}$ (mL/cm $H_2O$) | 35.7 [29.1–46.4] | 35.2 [29.1–46.5] | 37.5 [29.3–46.1] |
| Arterial blood gas analyses | | | |
| pH | 7.38 [7.31–7.45] | 7.37 [7.31–7.45] | 7.40 [7.33–7.46] |
| $pCO_2$ (kPa) | 5.3 [4.9–6.2] | 5.4 [4.9–6.1] | 5.2 [4.9–6.5] |
| Bicarbonate (mmol/L) | 25 [21–28] | 24 [21–27] | 25 [21–28] |
| $pO_2$ (kPa) | 9.9 [9.1–11.6] | 10.2 [9.2–12.1] | 9.8 [9.1–11.1] |
| $SaO_2$ (%) | 95 [94–97] | 95 [93–98] | 96 [94–97] |
| $PaO_2$ / $FiO_2$ ratio (mmHg) | 234 [176–289] | 242 [178–299] | 230 [172–278] |

Data are median [IQR], mean (SD) or n (%).

Abbreviations: PBW: predicted bodyweight; BMI: body mass index; APACHE: Acute Physiology and Chronic Health Evaluation; COPD: chronic obstructive pulmonary disease; COVID–19; coronavirus disease 2019; mL: milliliter; cm $H_2O$: centimeters of water; L: liter; sec: seconds; kPa: kilopascal; J/min: joule per minute; MP: mechanical power; $V_T$: tidal volume; RR: respiratory rate; Pmax: maximum airway pressure; PEEP: positive end–expiratory pressure; Pinsp: set inspiratory pressure; PS: set pressure support; ΔP: driving pressure; Tinsp: inspiratory time; $FiO_2$: fraction of inspired oxygen; $etCO_2$: end–tidal carbon dioxide; $SpO_2$: pulse oximetry; $C_{RS}$: compliance of the respiratory system; $pCO_2$: partial pressure of carbon dioxide; kPa: kilopascal; $pO_2$: partial pressure of oxygen; $SaO_2$: arterial oxygen saturation; mmHg: millimeters of mercury.

[+] available in passive patients

[*] available in active patients

## Ventilatory parameters

In the analysis including all patients, automated ventilation did not affect $V_T$, Pmax, PEEP, dynamic ΔP and Pinsp, but did result in a lower RR and minute volume (**Table 2**). $V_T$ increased with automated ventilation in passive patients, but not in active patients. In passive patients, RR, Pinsp, ΔP and minute volume decreased with automated ventilation, in active patients, RR and minute volume also decreased, but Pmax and ΔP increased (**Table 2**, and **S4** and **S5** **Figs**).

**Table 2. Ventilatory parameters.**

| | automated ventilation | conventional ventilation | mean difference (95% CI) | p |
|---|---|---|---|---|
| **All patients (n = 96)** | | | | |
| *Primary endpoint* | | | | |
| MP, median [IQR] and mean (SD) (J/min) | 15.8 [11.5–21.0] 17.3 (8.9) | 16.1 [10.9–22.6] 17.6 (8.7) | –0.44 (–1.17 to 0.29) | 0.24 |
| *Ventilation variables and parameters* | | | | |
| $V_{Ti}$ (mL) | 485 [402–585] | 469 [412–542] | 6.98 (–2.90 to 16.86) | 0.17 |
| $V_{Te}$ (mL) | 483 [410–581] | 483 [424–553] | 2.88 (–7.08 to 12.84) | 0.57 |
| $V_T$ (ml/kg IBW) | 7.2 [5.9–8.1] | 6.9 [6.1–8.0] | 0.07 (–0.08 to 0.22) | 0.34 |
| RR (breaths /minute) | 17 [14–22] | 18 [15–23] | –1.02 (–1.47 to –0.56) | < 0.01 |
| Minute volume (cm $H_2O$) | 8.2 [6.8–10.6] | 8.8 [7.5–10.4] | –0.40 (–0.65 to –0.14) | < 0.01 |
| Pmax (cm $H_2O$) | 21 [18–25] | 21 [17–25] | 0.18 (–0.24 to 0.60) | 0.40 |
| PEEP, set (cm $H_2O$) | 8 [6–11] | 8 [6–10] | 0.13 (–0.05 to 0.32) | 0.16 |
| $P_{insp}$ (cm $H_2O$)+ | 12 [10–15] | 13 [12–15] | –0.17 (–0.49 to 0.15) | 0.30 |
| PS (cm $H_2O$)* | 9 [6–13] | 8 [5–12] | 0.49 (0.04 to 0.95) | 0.03 |
| ΔP, dynamic (cm $H_2O$) | 12 [10–15] | 13 [10–16] | 0.05 (–0.31 to 0.40) | 0.80 |
| Flow (L/min) | 43.7 [36.4–52.3] | 44.5 [37.3–53.2] | –1.47 (–2.28 to –0.66) | < 0.01 |
| Tinsp (sec) | 1.09 [0.85–1.41] | 1.05 [0.84–1.24] | 0.11 (0.09 to 0.14) | < 0.01 |
| $FiO_2$ (%) | 31 [29–38] | 33 [28–40] | –0.64 (–1.36 to 0.09) | 0.09 |
| $etCO_2$ (kPa) | 4.9 [4.4–5.3] | 4.8 [4.3–5.4] | 0.09 (0.05 to 0.13) | < 0.01 |
| $SpO_2$ (%) | 94 [93–96] | 95 [93–97] | –0.38 (–0.61 to –0.15) | 0.01 |
| $C_{RS}$ (mL/cm $H_2O$) | 37.6 [30.4–49.9] | 36.9 [29.5–50.4] | –1.91 (–4.00 to 0.19) | 0.07 |
| *Arterial blood gas analyses results* | | | | |
| pH | 7.38 [7.31–7.44] | 7.40 [7.34–7.45] | –0.007 (–0.02 to 0.003) | 0.16 |
| $pCO_2$ (kPa) | 5.4 [4.9–9.1] | 5.4 [4.9–5.9] | 0.08 (–0.06 to 0.22) | 0.29 |
| Bicarbonate (mmol/L) | 24 [21–27] | 25 [21–28] | 0.10 (–0.30 to 0.50) | 0.62 |
| $pO_2$ (kPa) | 9.9 [9.1–11.6] | 9.9 [9.0–11.3] | 0.01 (–0.53 to 0.55) | 0.96 |
| $SaO_2$ (%) | 95 [93–97] | 96 [94–97] | –0.31 (–0.70 to 0.08) | 0.13 |
| $PaO_2$/$FiO_2$ ratio (mmHg) | 231 [181–299] | 229 [184–286] | –2.65 (–14.46 to 9.17) | 0.66 |
| **Passive patients (n = 45)** | | | | |
| *Primary endpoint* | | | | |
| MP, median [IQR] and mean (SD) (J/min) | 16.9 [12.5–22.1] 17.8 (7.2) | 19.0 [14.1–25.0] 19.4 (6.8) | –1.76 (–2.47 to –10.34) | < 0.01 |
| *Ventilation variables and parameters* | | | | |
| $V_{Ti}$ (mL) | 474 [401–571] | 462 [411–515] | 18.07 (7.88 to 28.24) | < 0.01 |
| $V_{Te}$ (mL) | 481 [404–578] | 466 [427–532] | 10.42 (.01 to 20.85) | 0.05 |
| $V_T$ (ml/kg PBW) | 7.0 [5.8–7.9] | 6.5 [5.8–7.6] | 0.22 (0.08 to 0.37) | < 0.01 |
| RR (breaths /minute) | 17 [13–22] | 18 [16–22] | –1.77 (–2.26 to –1.28) | < 0.01 |
| Minute volume (cm $H_2O$) | 7.9 [6.7–9.4] | 8.7 [7.7–9.8] | –0.61 (–0.86 to –0.37) | 0.01 |
| Pmax (cm $H_2O$) | 22 [19–26] | 22 [19–28] | –0.59 (–1.17 to –0.01) | 0.05 |
| Pplat (cm $H_2O$) | 19 [16–22] | 20 [16–23] | –0.19 (–0.57 to 0.19) | 0.33 |
| PEEP, set (cm $H_2O$) | 8 [6–11] | 8 [6–11] | 0.04 (–0.30 to 0.22) | 0.77 |
| PEEP, total (cm $H_2O$) | 8.7 [6.9–12] | 9.0 [7.0–12] | 0.19 (–0.08 to 0.46) | 0.17 |
| $P_{insp}$ (cm $H_2O$) | 13 [11–15] | 13 [12–15] | –0.38 (–0.69 to –0.07) | 0.02 |
| ΔP, static (cm$H_2O$) | 9 [8–11] | 10 [9–12] | –0.35 (–0.62 to –0.07) | < 0.01 |
| Flow (L/min) | 42.9 [36.2–48.4] | 44.2 [38.9–50.9] | –0.74 (–1.78 to 0.31) | 0.17 |
| Tinsp (sec) | 1.20 [0.94–1.60] | 1.11 [0.91–1.25] | 0.18 (0.14 to 0.22) | < 0.01 |
| $FiO_2$ (%) | 33 [30–40] | 35 [30–45] | –1.26 (–2.49 to –0.03) | 0.04 |
| $etCO_2$ (kPa) | 5.0 [4.6–5.3] | 4.8 [4.3–5.3] | 0.15 (0.09 to 0.21) | < 0.01 |
| $SpO_2$ (%) | 94 [93–96] | 95 [94–97] | –0.55 (–0.86 to –0.25) | < 0.01 |

*(Continued)*

**Table 2.** (Continued)

| | automated ventilation | conventional ventilation | mean difference (95% CI) | p |
|---|---|---|---|---|
| **All patients (n = 96)** | | | | |
| $C_{RS}$ (mL/cm $H_2O$) | 34.9 [29.2–42.5] | 32.9 [27.1–40.1] | 2.30 (0.38 to 4.22) | 0.02 |
| *Arterial blood gas analyses* | | | | |
| pH | 7.35 [7.29–7.39] | 7.38 [7.30–7.43] | −0.01 (−0.03 to 0.008) | 0.30 |
| $pCO_2$ (kPa) | 5.7 [5.1–64] | 5.4 [4.9–6.2] | 0.10 (−0.14 to 0.35) | 0.42 |
| Bicarbonate (mmol/L) | 24 [20–26] | 23 [21–26] | 0.30 (−0.34 to 0.95) | 0.37 |
| $pO_2$ (kPa) | 10.3 [9.2–12.4] | 10.0 [9.1–12.1] | 0.21 (−0.55 to 0.98) | 0.58 |
| $SaO_2$ (%) | 95 [93–98] | 96 [93–97] | −0.27 (−0.88 to 0.33) | 0.39 |
| $PaO_2/FiO_2$ ratio (mmHg) | 238 [182–305] | 220 [180–282] | 2.09 (−14.14 to 18.35) | 0.93 |
| **Active patients (n = 32)** | | | | |
| *Primary endpoint* | | | | |
| MP, median [IQR] and mean (SD) (J/min) | 14.6 [11.0–20.3] 16.3 (7.9) | 14.1 [10.1–21.3] 16.4 (8.9) | −0.81 (−2.13 to 0.49) | 0.23 |
| *Ventilation variables and parameters* | | | | |
| $V_{Ti}$ (mL) | 480 [402–573] | 485 [420–559] | −12.0 (−37.84 to 13.94) | 0.37 |
| $V_{Te}$ (mL) | 480 [415–563] | 499 [420–570] | −10.9 (−34.37 to 12.81) | 0.37 |
| $V_T$ (ml/kg PBW) | 7.2 [5.9–8.5] | 7.2 [6.4–8.3] | −0.19 (−0.58 to 0.21) | 0.36 |
| RR (breaths /minute) | 18 [15–22] | 19 [15–26] | −1.62 (−2.60 to −0.66) | 0.01 |
| Minute volume (cm $H_2O$) | 8.9 [7.1–11.3] | 9.5 [7.4–12.3] | −0.91 (−1.48 to −0.33) | 0.01 |
| Pmax (cm $H_2O$) | 20 [16–23] | 17 [14–22] | 1.21 (0.37 to 2.07) | 0.01 |
| PEEP, set (cm $H_2O$) | 8 [6–10] | 8 [5–10] | 0.42 (0.06 to 0.78) | 0.02 |
| PS (cm $H_2O$) | 9 [5–12] | 7 [5–12] | 0.47 (−0.15 to 1.09) | 0.13 |
| ΔP, dynamic (cm$H_2O$) | 11 [8–15] | 10 [7–14] | 0.81 (0.06 to 1.57) | 0.04 |
| Flow (L/min) | 42.5 [36.5–55.1] | 47.2 [37.5–58.2] | −4.29 (−6.23 to −2.36) | < 0.01 |
| $T_{insp}$ (sec) | 0.98 [0.81–1.13] | 0.95 [0.74–1.09] | 0.06 (0.02 to 0.10) | < 0.01 |
| $FiO_2$ (%) | 30 [26–37] | 30 [25–40] | −0.71 (−1.82 to 0.39) | 0.21 |
| $etCO_2$ (kPa) | 4.9 [4.3–5.4] | 4.9 [4.3–5.4] | 0.08 (0.01 to 0.15) | 0.03 |
| $SpO_2$ (%) | 94 [92–96] | 95 [93–97] | −0.59 (−1.21 to 0.04) | 0.07 |
| $C_{RS}$ (mL/cm $H_2O$ | 42.6 [30.8–57.2] | 50.7 [34.9–59.4] | −9.77 (−14.91 to −4.65) | 0.01 |
| *Arterial blood gas analyses results* | | | | |
| pH | 7.43 [7.39–7.46] | 7.44 [7.40–7.48] | −0.01 (−0.02 to 0.002) | 0.12 |
| $pCO_2$ (kPa) | 5.3 [4.6–5.9] | 5.1 [4.7–5.7] | 0.14 (−0.09 to 0.38) | 0.26 |
| Bicarbonate (mmol/L) | 27 [23–29] | 27 [23–31] | 0.01 (−0.72 to 7.37) | 0.98 |
| $pO_2$ (kPa) | 9.6 [8.9–10.7] | 9.5 [8.6–10.7] | 0.17 (−0.78 to 1.13) | 0.73 |
| $SaO_2$ (%) | 95 [94–97] | 96 [94–97] | −0.28 (−1.09 to 0.52) | 0.50 |
| $PaO_2/FiO_2$ ratio (mmHg) | 229 [175–270] | 244 [194–281] | −11.98 (−31.83 to 8.15) | 0.25 |

Data are median [IQR] or mean (SD)

mL: milliliter; cm $H_2O$: centimeters of water; L: liter; sec: seconds; kPa: kilopascal; J/min: joule per minute; MP: mechanical power; $V_T$: tidal volume; RR: respiratory rate; Pmax: maximum airway pressure; PEEP: positive end–expiratory pressure; Pinsp: set inspiratory pressure; PS: set pressure support; ΔP: driving pressure; $T_{insp}$: inspiratory time; $FiO_2$: fraction of inspired oxygen; $etCO_2$: end–tidal carbon dioxide; $SpO_2$: pulse oximetry; $C_{RS}$: compliance of the respiratory system; $pCO_2$: partial pressure of carbon dioxide; kPa: kilopascal; $pO_2$: partial pressure of oxygen; $SaO_2$: arterial oxygen saturation; mmHg: millimeters of mercury; 95% CI; 95% confidence interval.

[+] available in passive patients

[*] available in active patients

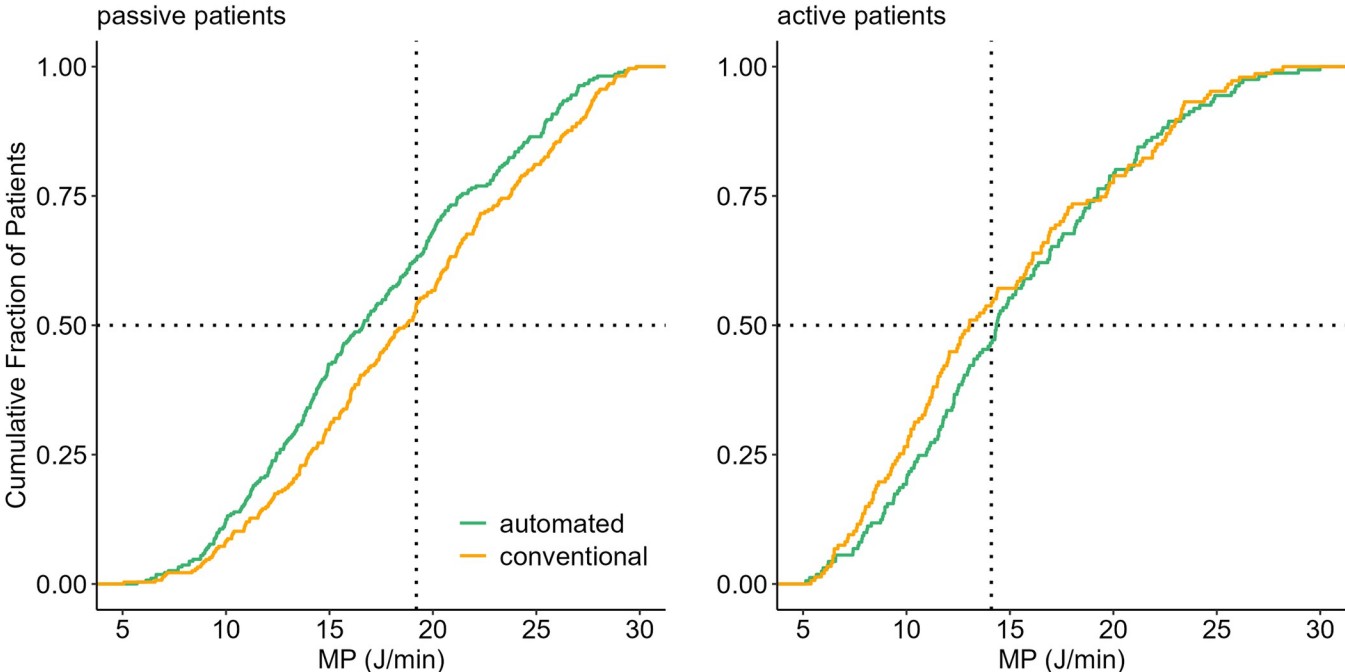

**Fig 2. Distribution plots of MP with automated ventilation and conventional ventilation in passive and active patients, showing all measurements of every patient.** Vertical dotted lines represent the median value with conventional ventilation. Horizontal dotted lines show the respective proportion of patients reaching each cutoff. Abbreviation: MP, mechanical power.

In passive patients, with use of the automated mode the proportions of MP > 17 J/min decreased, the proportion of patients with $V_T$ > 8 ml/kg PBW increased, and the proportions of patients with RR > 16, Pplat > 20 cmH$_2$O and flow > 45 L/min decreased (**Fig 3**). In active patients, with use of this mode MP > 17 J/min decreased, the proportion of patients with $V_T$ > 8 ml/kg PBW and Pmax >20 cmH$_2$O increased, and the proportions of patients with RR > 16 and flow > 45 L/min decreased.

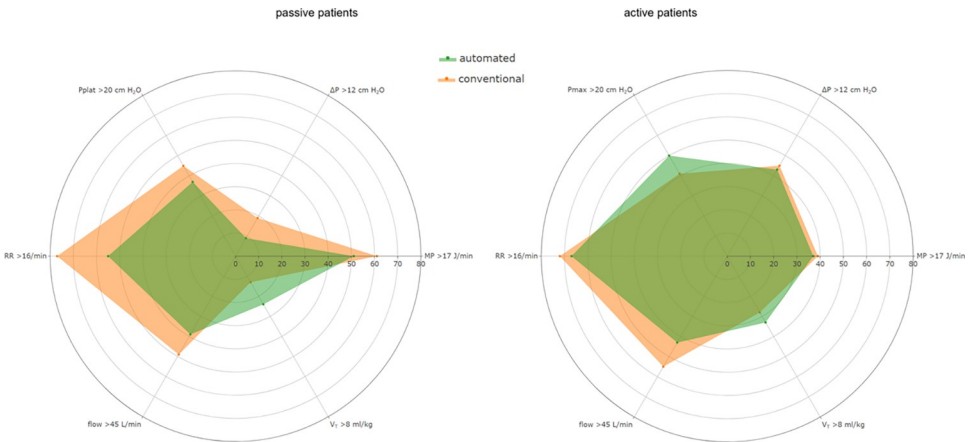

**Fig 3. Spider plots of MP, and the ventilatory parameters used in the calculation of mechanical power, with automated ventilation and conventional ventilation and in passive and in active patients.** Data is shown in percentages, of the proportion of patients, above the following cutoffs: 17 J/min for MP, 8 ml/kg for $V_T$, 45 L/min for flow, 16 breaths/min for RR, 20 cm H$_2$O for Pplat or Pmax and 12 cm H$_2$O for $\Delta$P. Abbreviations: MP, mechanical power; $V_T$, tidal volume; RR, respiratory rate; Pplat, plateau pressure; Pmax, maximum airway pressure; $\Delta$P, driving pressure.

## Posthoc analyses

Focusing on patients that received either pressure controlled or pressure support ventilation before randomization, the switch to automated ventilation resulted in a lower MP (**S1** and **S2 Tables**). MP did not change in patients that already received automated ventilation before randomization (**S3** and **S4 Tables**). Opposite to patients with a high $C_{RS}$, in patients with a low $C_{RS}$ MP was lower with automated ventilation (**S5**–**S7 Tables**). The proportion of time points with MP > 17 J/min and MP ≤ 17 J/min, and the time–weighted average comparison was not different between automated and conventional ventilation (**S8 Table**).

## Discussion

The findings of this international, multicenter, randomized crossover clinical trial that compared a commercially available automated ventilation mode to conventional ventilation with respect to MP in critically ill invasively ventilated patients can be summarized as follows: (1) INTELLiVENT–ASV does reduce MP, albeit only in passive patients. The reduction in MP was associated with (2) a decrease in RR; and (3) an increase in $V_T$.

Our study has the following strengths. We used a crossover design, meaning that each patient served as his or her own control, increasing the statistical power of the investigation. We performed the study in two academic and two non–academic ICUs in two countries, increasing the generalizability of the findings. ICU nurses and doctors in these ICUs were all experienced with the here tested automated ventilation mode, but also the use of lung–protective ventilation with conventional ventilation. We collected ventilation parameters with inspiratory and expiratory holds, increasing the reliability of the calculation of MP. Last but not least, the study protocol and the statistical analysis plan were strictly followed at all times.

To the best of our knowledge, this is the first randomized cross–over study that determined the effects of automated ventilation by means of INTELLiVENT–ASV on MP in an unselected critically ill ICU population. Previous research has suggested benefits in terms of lower MP with INTELLiVENT–ASV, but most studies had an observational design [21–23]. Two previous studies specifically investigated the effect of INTELLiVENT–ASV compared to conventional ventilation in patients with coronavirus disease 2019 [21, 22]. These studies demonstrated a reduction in MP with INTELLiVENT–ASV, primarily driven by a lower RR. Another study assessed whether INTELLiVENT–ASV could provide lung–protective settings, resulting in lower ΔP and MP [23]. Additionally, a small RCT focused on ARDS patients [24]. Although the primary outcome was the transpulmonary ΔP, that study found lower transpulmonary MP with INTELLiVENT–ASV due to a lower RR at the expense of a larger $V_T$. Similarly, in patients after cardiothoracic surgery [30], a lower MP was reported with the use of INTELLiVENT–ASV. Our findings confirm those of the previous investigations, indicating that INTELLiVENT–ASV has the potential to adjust ventilator settings in a manner that reduces MP.

One major finding in our study was that the automated ventilation mode lowers MP only in passive patients, and we establish a correlation between the decline in MP and a decrease in RR. In passive patients, $V_T$ and RR are under full control of the automated ventilation algorithms, which target the lowest work of breathing (the 'Otis equation') [19] and the lowest force of breathing (the 'Mead equation') (the 'Mead equation') [20] resulting in a lower RR and a higher $V_T$. This is in line with the findings of the abovementioned study in patients with severe acute hypoxemic respiratory failure due to coronavirus disease 2019 [21].

In actively breathing patients, MP was not affected by INTELLiVENT–ASV. In active patients, specifically those with a high respiratory drive, the tested automated ventilation mode can only increase the pressure support level in an attempt to increased $V_T$, and consequently a lower RR. Of note, MP is low in active compared to passive patients. Active patients use their

diaphragm promoting alveolar recruitment [31]. With this, total lung volume increases and airway pressures decline, leading to a lower MP. It should be noted, though, that there is an ongoing debate on whether MP can be calculated in active patients [32], and herein how to measure breathing effort [33]––the equation we used for calculating MP can easily lead to an underestimation of MP in active patients [34–36].

If patients received conventional ventilation before randomization, the switch to the automated mode resulted in a lower MP. In patients that were already receiving automated ventilation before randomization, MP was not affected. Probably in these last patients, INTELLiVENT–ASV had already optimized the ventilator settings [37], settings that may have been taken over when starting the first study phase. The findings of the posthoc analysis in which we compared the effect of automated ventilation on MP in patients with low and relatively normal $C_{RS}$ showed a decrease in MP only in patients with a poor lung condition. We cannot exclude that this finding was driven by the higher proportion of active patients in the group of patients with a relative normal $C_{RS}$.

In clinical practice, it can be challenging to set the ventilator so that ventilation is applied with a low MP [38, 39], in particular because it is uncertain which ventilator setting to prioritize herein [40]. For example, choosing a lower $V_T$ may cause a reduction in MP [2], but if a compensatory higher RR is needed, any effect on MP could be nullified, or worse. Indeed, an increase in RR will increase MP [34]. Our study shows how an automated mode of ventilation that targets a low work–and force of breathing automatically sets ventilator settings so that MP is reduced. No manual interventions by ICU nurses or doctors were needed.

Thus far, lung–protective ventilation has focused on the use of a correct, i.e., low $V_T$ [12], and less on the potential injurious effects of the RR [41]. One salient finding in our study was that with the tested automated ventilation mode $V_T$ actually increases, and RR decreases. Recent studies show that ventilation strategies with use of low $V_T$ and higher RR may only be beneficial in patients with low $C_{RS}$ [17, 42], while it results in harm in patients with a relative normal $C_{RS}$. Interestingly, in patients with a low $C_{RS}$ the automated ventilation mode chooses a lower RR and a higher $V_T$, resulting in less MP. Future studies are needed to understand whether this reduction in MP [32], achieved by a reduction in RR and an increase in $V_T$ translates in better outcomes [43], such as duration of ventilation and maybe even mortality [44].

Our study has limitations. The two crossover phases of the study were relatively short in time, and it could be that the effects of the tested ventilation mode are different if applied for a longer duration. Most patients were hemodynamic and respiratory stable, hampering translation to unstable patients. Patients were included early after initiation of the ventilation, while the effect of the tested ventilation mode could be different at later phases. Due to the nature of the intervention, ICU professionals taking care of the patients and researchers collecting the data could not be blinded. However, the study protocol was strictly followed, and all preplanned analyses were performed by investigators that remained blinded for the allocated ventilation mode.

## Conclusion

In this cohort of critically ill patients expected to need invasive ventilation for > 24 hours, automated ventilation by means of INTELLiVENT–ASV did not reduce MP. In passive patients, automated ventilation reduced MP. In these patients, the reduction seems mainly driven by a reduction in RR, and happens despite an increase in $V_T$.

## Supporting information

**S1 Checklist. CONSORT checklist.**
(PDF)

**S1 File. Study protocol.**
(PDF)

**S1 Fig. Figures of the residual analysis of all patients figures of the residual analysis, of mixed-effect generalized linear model for the primary endpoint, including all patients.**
(DOCX)

**S2 Fig. Figures of the residual analysis of passive patients figures of the residual analysis, of the mixed-effect generalized linear model including only passive patients.**
(DOCX)

**S3 Fig. Figures of the residual analysis of active patients figures of the residual analysis, of the mixed-effect generalized linear model including only active patients.**
(DOCX)

**S4 Fig. Distribution plots of ventilatory parameters in passive patients.** Distribution plots of ventilatory parameters in passive patients, showing all measurements of every patient. Vertical dotted lines represent the median value with conventional ventilation. Horizontal dotted lines show the respective proportion of patients reaching each cutoff.
(DOCX)

**S5 Fig. Distribution plots of ventilatory parameters in active patients.** Distribution plots of ventilatory parameters in active patients, showing all measurements of every patient. Vertical dotted lines represent the median value with conventional ventilation. Horizontal dotted lines show the respective proportion of patients reaching each cutoff.
(DOCX)

**S1 Table. Ventilatory parameters, conventional ventilation.** Ventilatory parameters, conventional ventilation before randomization (n = 34).
(DOCX)

**S2 Table. Ventilatory parameters in passive patients, conventional ventilation.** Ventilatory parameters in passive patients, conventional ventilation before randomization.
(DOCX)

**S3 Table. Ventilatory parameters, automated ventilation.** Ventilatory parameters, automated ventilation before randomization (n = 62).
(DOCX)

**S4 Table. Ventilatory parameters in passive patients, automated ventilation.** Ventilatory parameters in passive patients, automated ventilation before randomization (n = 56).
(DOCX)

**S5 Table. Ventilatory parameters with $C_{RS} \leq 37.3$.** Ventilatory parameters in patients with $C_{RS} \leq 37.3$ (n = 50).
(DOCX)

**S6 Table. Ventilatory parameters in passive patients with $C_{RS} \leq 37.3$.** Ventilatory parameters in passive patients with $C_{RS} \leq 37.3$ (n = 32).
(DOCX)

**S7 Table. Ventilatory parameters in patients with CRS > 37.3.** Ventilatory parameters in patients with $C_{RS} > 37.3$ (n = 46).
(DOCX)

**S8 Table. Proportion of time points and time weighted average.** Results of the proportion of time points and time weighted average analysis.
(DOCX)

## Acknowledgments

**Collaborative authors (in alphabetic order):**

Rik J.A. Appel, Amanda van den Berg, Marjolein C.W.M. Bierlee, Ben H. de Boer, Danique Boezaart, José A. Boots, Michela Botta, Bibi Bosman, Philippe Bühler, Laura A. Buiteman–Kruizinga, Kim–Jana Fehlbier, Robin L. Goossen, Anastasia A. Guseva, Pim L.J. van der Heiden, Coby Hoekstra–Kapitein, Eva–Maria Kleinert, Hans Last, Tobias D. van Leijsen, Stephanie S. List, Marieke Luttmer–Laven, Frederique Paulus, Lotte Remmerswaal, Yvonne Schriel–van den Berg, Marcus J. Schultz, Ary Serpa Neto, Jante S. Sinnige, Anissa M. Tsonas, Patricia van Velzen, Tom Vermeulen, Pedro Wendel Garcia

## Author Contributions

**Conceptualization:** Laura A. Buiteman-Kruizinga, Marcus J. Schultz, Pim L. J. van der Heiden, Frederique Paulus.

**Data curation:** Laura A. Buiteman-Kruizinga, Michela Botta, Stephanie S. List, Ben H. de Boer, Patricia van Velzen, Philipp Karl Bühler, Pedro D. Wendel Garcia, Pim L. J. van der Heiden, Frederique Paulus.

**Formal analysis:** Laura A. Buiteman-Kruizinga, Ary Serpa Neto, Marcus J. Schultz.

**Investigation:** Laura A. Buiteman-Kruizinga, Marcus J. Schultz, Frederique Paulus.

**Methodology:** Laura A. Buiteman-Kruizinga, Ary Serpa Neto, Marcus J. Schultz, Frederique Paulus.

**Project administration:** Laura A. Buiteman-Kruizinga, Stephanie S. List, Ben H. de Boer, Patricia van Velzen, Philipp Karl Bühler, Pedro D. Wendel Garcia, Marcus J. Schultz, Pim L. J. van der Heiden, Frederique Paulus.

**Supervision:** Marcus J. Schultz, Pim L. J. van der Heiden, Frederique Paulus.

**Validation:** Laura A. Buiteman-Kruizinga.

**Visualization:** Laura A. Buiteman-Kruizinga, Pim L. J. van der Heiden.

**Writing – original draft:** Laura A. Buiteman-Kruizinga, Marcus J. Schultz, Pim L. J. van der Heiden, Frederique Paulus.

**Writing – review & editing:** Ary Serpa Neto, Michela Botta, Stephanie S. List, Ben H. de Boer, Patricia van Velzen, Philipp Karl Bühler, Pedro D. Wendel Garcia, Marcus J. Schultz, Pim L. J. van der Heiden, Frederique Paulus.

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
