## [Decision Letter · Decision Letter 0]

25 Mar 2024

PONE-D-24-01411Effect of Automated versus Conventional Ventilation on Mechanical Power of Ventilation – a randomized crossover clinical trialPLOS ONE

Dear Dr. Buiteman-Kruizinga,

Thank you for submitting your manuscript to PLOS ONE. After careful consideration, we feel that it has merit but does not fully meet PLOS ONE’s publication criteria as it currently stands. Therefore, we invite you to submit a revised version of the manuscript that addresses the points raised during the review process.

We look forward to receiving your revised manuscript.

Kind regards,

Samuele Ceruti

Academic Editor

PLOS ONE

Journal Requirements:

"LBK received fees from Hamilton Medical for lecturing. MJS was part-time employed as a team leader of Research and New Technologies at Hamilton Medical from January 2022 till January 2023. The other authors declare no conflicts of interest."

3. In the online submission form, you indicated that "data available on reasonable request including a statistical analysis plan"

**Additional Editor Comments:**

ACADEMIC EDITOR:The study is interesting and could have a good appeal; however, it is necessary to be more rigorous in the statistical and methodological part in general, not only evading what the reviewers point out, but also sticking more closely to the elements that are highlighted in the results part. It is an interesting paper, but still far from being resolved; resolving the problems highlighted by the reviewers does not yet guarantee publication of the paper.

Reviewers' comments:

Reviewer's Responses to Questions

**Comments to the Author**

1. Is the manuscript technically sound, and do the data support the conclusions?

Reviewer #1: Yes

Reviewer #2: Partly

2. Has the statistical analysis been performed appropriately and rigorously? 

Reviewer #1: Yes

Reviewer #2: No

3. Have the authors made all data underlying the findings in their manuscript fully available?

Reviewer #1: Yes

Reviewer #2: No

4. Is the manuscript presented in an intelligible fashion and written in standard English?

Reviewer #1: Yes

Reviewer #2: Yes

5. Review Comments to the Author

Reviewer #1: First, I thank the publisher for allowing me to review this manuscript.

The authors in this work wanted to compare the intellivent-ASV with conventional ventilation by evaluating the "behavior", "trend" of PD in passive and active patients and secondly which ventilator settings influenced PD. The authors test the hypothesis that automated mechanical ventilation results in lower PD.

Abstract

I advise authors, first of all, to remove abbreviations in the abstract. This ensures that the abstract is more fluid and readable. Always remaining within the limits permitted by the magazine.

Keyword

Clear and precise. Suitable for the work and the subject of the study. Unfortunately, many chosen keywords are not included in the MESH database, I advise authors to balance the keywords not present with those present in order to facilitate indexing of the manuscript in case of publication.

Introduction

Materials and methods

I advise the authors in the study intervention section to describe how the INTELLIVENT reasons and works, since otherwise it is not clear why SpO2 and EtCO2 and the other values are set on the ventilator. The authors must think about the fact that not everyone knows this type of ventilation just as not everyone knows ASV ventilation. In some part of the manuscript, or in the introduction or better yet in a methods section, the authors must think of introducing and describing both the ASV and the INTELLIVENT which is its evolution.

Question for the authors. When clinicians set up "conventional" ventilation, do they always use Hamilton ventilators or any ventilator present in the departments? Perhaps the authors should specify it, for completeness of information.

Discussion

I advise authors to engage a little more with the existing literature. In the discussion elaborated now there is a detailed and adequate explanation of the results with different hypotheses on some results, without, however, a real comparison with the present literature. References appear every now and then but they seem almost forced. This section needs to be better renovated.

Reviewer #2: The manuscript addresses an interesting topic. The data are original and the methods sound in general. Nevertheless, there are several aspects deserving clarifications and in depth descriptions. Some comments follow.

1. Please, provide the data and the code used to obtain the results discussed in paper. This ensures the reproducibility of the results. It would be a plus with no doubts.

2. The employed statistical methods are sound. I really appreciate the use of mixed models. Nevertheless, there are several aspects which should be carefully checked. Firstly, the Gauss-Markov assumptions must be met; please, provide an extensive residual analysis confirming that the linear model is suitable for the data at hand and, if skewness, outliers, etc arise, please modify the model accordingly. Secondly, the data are not longitudinal but also multilevel, as multiple centers are considered: please, add random effects at the center level too. As far I can see, the quantile (median) regression with random effects could be a practical and useful alternative, as long as the residuals do not look Gaussian-distributed.

3. As the number of predictors and confounders is high, please provide guidance towards variables selection in the mixed effect regression framework. Please, bear in mind that it is of fundamental importance to keep confounders (mainly demographic variables) in the model to ensure that differences in the outcome do not arise because of omitted variables and heterogeneity in general.

4. About missingness, the authors state "We made no assumption for missing data". I guess that a missing completely at random mechanism is considered, am I right? I strongly suggest to investigate the role of missingness, as missing not at random may strongly bias the results. Indeed, dating back to the taxonomy introduced by Rubin, it is well known that ignoring the role played by missing values may strongly affect statistical inference. I suggest to look at pattern mixture approaches, as they are rather simple to be implemented.

5. Even for simple t-tests, please check the underlying assumptions and provide evidence (in a supplement) that inference is valid.

6. PLOS authors have the option to publish the peer review history of their article (what does this mean?). If published, this will include your full peer review and any attached files.

Reviewer #1: No

Reviewer #2: No

---

## [Author Response · Author response to Decision Letter 0]

30 Apr 2024

response is added at the documents in attach files

---

## [Decision Letter · Decision Letter 1]

2 Jul 2024

Effect of Automated versus Conventional Ventilation on Mechanical Power of Ventilation – a randomized crossover clinical trial

PONE-D-24-01411R1

Dear Dr. Buiteman-Kruizinga,

We’re pleased to inform you that your manuscript has been judged scientifically suitable for publication and will be formally accepted for publication once it meets all outstanding technical requirements.

Kind regards,

Samuele Ceruti

Academic Editor

PLOS ONE

Additional Editor Comments (optional):

Reviewers' comments:

Reviewer's Responses to Questions

**Comments to the Author**

1. If the authors have adequately addressed your comments raised in a previous round of review and you feel that this manuscript is now acceptable for publication, you may indicate that here to bypass the “Comments to the Author” section, enter your conflict of interest statement in the “Confidential to Editor” section, and submit your "Accept" recommendation.

Reviewer #1: All comments have been addressed

Reviewer #2: All comments have been addressed

2. Is the manuscript technically sound, and do the data support the conclusions?

Reviewer #1: (No Response)

Reviewer #2: (No Response)

3. Has the statistical analysis been performed appropriately and rigorously? 

Reviewer #1: (No Response)

Reviewer #2: (No Response)

4. Have the authors made all data underlying the findings in their manuscript fully available?

Reviewer #1: (No Response)

Reviewer #2: (No Response)

5. Is the manuscript presented in an intelligible fashion and written in standard English?

Reviewer #1: (No Response)

Reviewer #2: (No Response)

6. Review Comments to the Author

Reviewer #1: (No Response)

Reviewer #2: (No Response)

7. PLOS authors have the option to publish the peer review history of their article (what does this mean?). If published, this will include your full peer review and any attached files.

Reviewer #1: No

Reviewer #2: No

---

## [Editor Report · Acceptance letter]

19 Jul 2024

PONE-D-24-01411R1 

PLOS ONE

Dear Dr. Buiteman-Kruizinga, 

I'm pleased to inform you that your manuscript has been deemed suitable for publication in PLOS ONE. Congratulations! Your manuscript is now being handed over to our production team.

Kind regards, 

on behalf of

Dr. Samuele Ceruti 

Academic Editor

PLOS ONE